# Integrating Metabolomics and Gut Microbiota to Identify Key Biomarkers and Regulatory Pathways Underlying Metabolic Heterogeneity in Childhood Obesity

**DOI:** 10.3390/nu17111876

**Published:** 2025-05-30

**Authors:** Zhiwei Xia, Yan Li, Jiyong Yin, Zhaolong Gong, Jing Sun, Shi Shen, Yi Yang, Tingting Liu, Liyuan Wang, Junsheng Huo

**Affiliations:** 1NHC Key Laboratory of Public Nutrition and Health, National Institute of Nutrition and Health, Chinese Center for Disease Control and Prevention, Beijing 100050, Chinayinjy@ninh.chinacdc.cn (J.Y.); wangly@ninh.chinacdc.cn (L.W.); 2Department of School Health, Beijing Center for Disease Control and Prevention, Beijing 100013, China; 3Key Laboratory of Diagnostic and Traceability Technologies for Food Poisoning, Beijing Center for Disease Control and Prevention, Beijing 100013, China

**Keywords:** childhood obesity, multi-omics, metabolic heterogeneity, gut microbiota, metabolomics

## Abstract

Background/Objectives: Individuals with childhood obesity exhibit significant metabolic heterogeneity, necessitating precise biomarkers for risk stratification and assessment. This multi-omics investigation characterizes metabolic and microbial signatures underlying divergent metabolic phenotypes in the context of pediatric obesity. Methods: We analyzed 285 Chinese children (5–7 years) stratified into five groups: wasting (WAS, *n* = 55), metabolically healthy/unhealthy and normal weight (MHWH, *n* = 54; MUWH, *n* = 67), and metabolically healthy/unhealthy obesity (MHO, *n* = 36; MUO, *n* = 73). Untargeted metabolomics (Orbitrap ID-X Tribrid™) and 16S rRNA sequencing were integrated with multivariate analyses (OPLS-DA with VIP > 1, FDR < 0.05; Maaslin 2 with TSS normalization and BH correction, FDR < 0.10). Results: Analysis identified 225 differential metabolites and 12 bacterial genera. The proportion of steroids and their derivatives among differential metabolites in the MUO/MHO group was significantly lower than that in the OVOB/NOR and OVOB/WAS groups (2.12% vs. 7.9–14.1%). MUO displayed elevated C17 sphinganine and LysoPC (O-18:0) levels but reduced PI (16:0/14:1) levels. In contrast, OVOB showed upregulated glycerol phospholipids (LPCs and PSs) and downregulated PE species (e.g., PE(16:0/16:0)) as well as gut microbiota dysbiosis characterized by a higher Firmicutes/Bacteroidetes (F/B) ratio (2.07 vs. 1.24 in controls, *p* = 0.009) and reduced α diversity (Ace index, Chao1 index, and Shannon index values were lower in the OVOB group, Shannon index: 2.96 vs. 3.45, *p* = 0.03). SCFA-producing genera were negatively correlated with the OVOB group, while positively associated with PE(16:0/16:0). Internal validation showed differential metabolites had potential predictive efficacy for MUO/MHO (AUC = 0.967) and OVOB/NOR (AUC = 0.888). Conclusions: We identified distinct lipid disruptions characterizing obesity subtypes, including steroid/terpene deficits and sphingolipid/ether lipid dysregulation in the MUO/MHO groups as well as phospholipid imbalance (↑LPC/PS↓PE) in the OVOB/NOR groups. The gut microbiota exhibited a profile characterized by low diversity, an increased F/B ratio, and a reduced abundance of SCFA-producing genera. These findings suggest potential biomarkers for childhood obesity stratification, though further validation is warranted.

## 1. Introduction

### 1.1. Global Burden and Clinical Significance of Childhood Obesity

Obesity has emerged as a major public health crisis, significantly increasing the risk of chronic diseases such as cardiovascular disorders, type 2 diabetes, and metabolic syndrome, thereby imposing substantial economic and societal burdens [1,2]. In particular, childhood obesity is alarmingly prevalent, with global rates quadrupling since 1990, affecting approximately 159 million children and adolescents in 2022 [1]. In China, the obesity rate among school-aged children surged from 0.1% in 1985 to 7.9% in 2020 [3,4], with projections exceeding 15% by 2030 [5]. Beyond its role as an independent risk factor for cardiometabolic diseases, obesity exhibits marked metabolic heterogeneity, which critically influences clinical outcomes [6]. For instance, children with metabolically unhealthy obesity (MUO) face a 3- to 5-fold higher risk of cardiovascular events compared to those with metabolically healthy obesity (MHO) [7]. However, conventional metrics such as body mass index (BMI) and popular biomarkers (e.g., blood lipids and insulin resistance indices) lack the sensitivity to distinguish these phenotypes [8]. These methods exhibit certain limitations in early monitoring and warning, risk assessment, and providing insights into specific molecular and biochemical changes during the development of obesity [9]. Consequently, these limitations hinder classified management as well as precise prevention and control of obesity [10].

### 1.2. Metabolic Heterogeneity in Pediatric Obesity: Challenges in Phenotyping

MHO refers to individuals who reach the obesity criteria but do not exhibit adverse metabolic effects, including IR, impaired glucose tolerance, dyslipidemia, and hypertension [7,11]. The transition from “metabolically healthy” to “unhealthy” states in pediatric obesity is driven by multifaceted interactions among genetic predisposition, epigenetic modifications, and environmental exposures [12]. Although adult obesity studies have identified branched-chain amino acids (BCAAs), aromatic amino acids (AAAs), and acylcarnitines as key obesity risk factors [13,14], these findings cannot be directly extrapolated to children due to developmental dynamics. One study reported that further metabolic pathway analysis of fatty acid biosynthesis, phenylalanine metabolism, and leucine and valine degradation can help to distinguish differences between childhood MHO and MUO individuals [15]. A study on the lipid profile characteristics of Danish children and adolescents showed that 142 lipid types showed significant differences in obese children; in particular, sphingolipids (such as ceramide) and PE (phosphatidylethanolamine) were significantly increased, whereas lysophospholipids (such as LysoPE and LysoPC) were significantly decreased in the obesity group [16]. However, these results were not observed in all populations of children due to different races, regions, ages, and lifestyle influences such as diet and exercise [17]. In addition, the diagnostic criteria of MHO and MUO in obese children are controversial, and simple indicators such as blood pressure, lipid, blood glucose, and waist circumference cannot accurately distinguish their metabolic differences and subsequent complications [11]. Such limitations highlight the necessity of pediatric-specific multi-omics investigations to elucidate metabolic pathways unique to childhood obesity.

### 1.3. Gut Microbiota in Obesity: Current Evidence and Controversies

The gut microbiota constantly interacts with the host and plays an important role in maintaining the host’s energy metabolism balance and immunity [18]. Over the past 5 years, numerous cross-sectional studies have shown that the gut microbiota of obese children is characterized by an increase in the relative abundance of Firmicutes, a decrease in the abundance of Bacteroidetes, an increase in the abundance of opportunistic pathogens, and a decrease in the abundance of probiotics [19,20]. However, studies and observations that dispute the above results have also been reported. For example, Chen compared the fecal flora of 28 obese children and 23 healthy children and did not observe a significant difference in the ratio of Firmicutes to Bacteroides (F/B) between the two groups [21]. Existing studies have not reached a consensus on the characteristics of the intestinal flora unique to obese children. Challenges remain, including population heterogeneity, an excessive focus on the differences in dominant bacteria genera, and a lack of dedicated databases. For example, *Akkerman mucophilic* has been identified as part of the next generation of probiotics that are highly beneficial to human metabolic health, despite its low proportion in the overall flora [22].

### 1.4. Objective of This Study

Childhood obesity combined with metabolic heterogeneous phenotypes shows different characteristics of gut microbiota composition and serum metabolomic profiles [7,23]. Studies have shown that the gut microbiota and its metabolites can interact with the enteric nervous system and the central nervous system in a bidirectional manner through two pathways: enteroendocrine and neurosecretion [18,24,25]. This bidirectional pattern is called the gut–brain axis [26]. The metabolic active products of the gut microbiota, such as short-chain fatty acids (SCFAs), bile acids, and acidic metabolites, as well as vitamin K, vitamin B12, biotin, and folic acid, affect hypothalamic inflammation, satiety signals, energy expenditure, and food intake through the gut–brain axis, thereby establishing a connection with the occurrence and development of obesity [18,27]. Meanwhile, lifestyle behaviors related to obesity, such as diet and exercise, can directly act on the gut microbiota. One study reported that a high-fat or high-carbohydrate diet induces an inflammatory response at the base of the hypothalamus through the gut–brain axis, causing central resistance to leptin (an appetite-regulating hormone) [28].

Therefore, this study conducted an untargeted metabolomic and 16S rRNA sequencing analysis of 285 Chinese children aged 5–7 years to uncover multiple omics features and potential biomarkers of childhood obesity and its associated metabolic phenotypes. This study also sought to explore possible biological pathways leading to obesity and abnormal metabolic phenotypes. We conducted a correlation analysis between differential metabolites and differential bacterial genera to explore possible metabolic or regulatory networks. In addition, this study not only focuses on the overall characteristics of obese children but also further subdivides metabolic phenotypes (such as MHO and MUO) to identify the metabolic and microbial differences between different phenotypes. This study identifies the significant differences between MUO and MHO in terms of metabolic pathways (lipid metabolism and sphingoid metabolism) and key metabolites (LysoPC and PE), highlighting the correlations between key differentiator genera and serum metabolites. We hypothesize that different obese children exhibit different characteristics of the gut microbiota and serum metabolites (for example, a reduction in the taxonomic groups that produce SCFA and dysregulation of lipid metabolites), and the interaction between bacterial genera and metabolites could be observed.

## 2. Materials and Methods

### 2.1. Study Design and Participants

This cross-sectional study utilized data from “Nutrition and Health Influencing Factors Cohort Survey and Intervention” (Project No. 102393220020070000016). Participants were recruited from a population of 1078 children aged 5–7 years in 2023. Anthropometric measurements (height and weight), dietary behaviors, and biological samples (blood and feces) were collected using standardized questionnaires.

Nutritional status was classified according to the 2007 WHO growth standards using age- and sex-specific BMI Z-scores (BMIZ) as follows: wasting (WAS), BMIZ < −2 SD (*n* = 55); overweight/obesity (OVOB), BMIZ > +1 SD (*n* = 109); and normal weight (NOR), BMIZ between −2 SD and +1 SD (*n* = 121, matched according to gender and age). Exclusion criteria included the following: (1) incomplete blood samples or unqualified specimens (hemolysis and chylous blood); (2) antibiotic/probiotic use within three months prior to enrollment; (3) diarrhea, constipation, and fever or diseases such as pediatric diarrhea and pediatric constipation diagnosed by a doctor within the past two weeks; and (4) severe clinically diagnosed metabolic diseases or neuroendocrine diseases (e.g., diabetes and congenital hypothyroidism). Metabolic phenotypes were further stratified based on consensus definitions [29], including triglycerides (TG) ≤ 1.7 mmol/L, high-density lipoprotein cholesterol (HDL-C) ≥ 1.03 mmol/L, fasting blood glucose (FBG) < 5.6 mmol/L, and systolic and diastolic blood pressure (SBP: systolic blood pressure; DBP: diastolic blood pressure) ≤ 90th percentile. When all criteria were met, the individual was considered to have a normal metabolic phenotype. Individuals with one unsatisfactory item were further classified as having an abnormal metabolic phenotype. The final groups included metabolically healthy and weight healthy (MHWH, *n* = 54); metabolically unhealthy and weight healthy (MUWH, *n* = 67); metabolically healthy obesity (MHO, *n* = 36); and metabolically unhealthy obesity (MUO, *n* = 73). Fecal samples were collected from a subset (*n* = 42) matched by age, sex, and BMIZ (OVOB (*n* = 13), WAS (*n* = 8), and NOR (*n* = 21)).

All procedures were approved by the Ethics Committee of the National Institute of Nutrition and Health, Chinese Center for Disease Control and Prevention (Approval No. 2018-017), with written informed consent obtained from legal guardians. The research flow design chart is shown in Figure 1.

### 2.2. Questionnaire Survey

A self-designed questionnaire was used to collect demographic characteristics (age, sex, parental occupation/education level, primary caregiver status, and annual household income), perinatal factors (birth weight, birth length, breastfeeding duration, formula feeding, and micronutrient supplementation), semi-quantitative food frequency questionnaire (FFQ), health status survey (antibiotic/probiotic use in the past three months; diseases and related symptoms, such as diarrhea, constipation, and fever in the past two weeks), and parents’ attitude toward nutrition knowledge. The questionnaire was completed by the child and the guardian. The on-site quality control officer checked the logic and completeness, and the errors or missing items were corrected immediately.

### 2.3. Anthropometry

Certified technicians performed measurements in compliance with WS/T 424-2013 national standards [30]. A height measuring instrument (accuracy 0.1 cm) and electronic weight scale (accuracy 0.05 kg), which were verified by the quality inspection department, were used to measure height and weight. Measurements were performed in a quiet, spacious, level, and solid environment according to the requirements of pediatric hygiene. Blood pressure was measured in accordance with the American Heart Association pediatric guidelines (Flynn et al., 2017 [31]). After a 10 min rest in a seated position, three consecutive readings were obtained from the right arm using a validated oscillometric device (OMRON HEM-7121, OMRON Healthcare Co., Ltd.,Kyoto, Japan) with appropriate cuff sizes (covering 80–100% of the arm circumference). The average of the last two measurements was recorded for analysis.

### 2.4. Biological Sample Collection

Sample collection and processing: (1) Blood sample: A 3 mL sample of venous blood from the survey subjects was collected using vacuum blood collection tubes, and the blood was inverted 5 times immediately after blood collection. After leaving at room temperature for 20–30 min, the sample was centrifuged at 3000 RPM for 15 min to separate the serum. The serum was transferred to a serum freezing tube for temporary storage at −20 °C and then transported to −80 °C for long-term storage. (2) Fecal samples: A sterile fecal collection tube with spoon and fecal protection solution were used to collect the subjects’ first stool in the morning. A spoonful of central inner stool (about 2 g) free of water and urine contamination was obtained using the plastic spoon. After adding the fecal protection solution, it was temporarily stored at −20 °C and transported to −80 °C for storage. All samples were tested for serum metabolomics and fecal 16S rRNA in the same batch.

### 2.5. Biochemical and Micronutrient Measurements

Serum biochemical parameters, including total cholesterol (TC), TG, HDL-C, low-density lipoprotein cholesterol (LDL-C), FBG, alanine aminotransferase (ALT), aspartate aminotransferase (AST), high-sensitivity C-reactive protein (CRP-H), adenosine diphosphate (ADP), and minerals (calcium, Ca; iron, Fe; and zinc, Zn), were analyzed using a Hitachi 7020 Automatic Biochemical Analyzer (Hitachi High-Technologies Corporation, Tokyo, Japan) with commercial assay kits (Leadman Biochemistry Co., Ltd., Beijing, China). Two-level quality controls (Leadman Biochemistry Co., Ltd., Beijing, China) were analyzed daily.

According to the “WS/T 553-2017” [32] and “WS/T 677-2020” [33] methods, serum vitamin A and 25(OH)D2/D3 levels were determined using high-performance liquid chromatography mass spectrometry (HPLC-MS/MS: Waters Corporation, Milford, MA, USA). Quantitative determination was performed using the isotope internal standard method.

### 2.6. Nontargeted Metabolomic Detection

#### 2.6.1. Reagents and Instrumentation

Mass spectrum-grade solvents (methanol and acetonitrile) and formic acid were procured from Fisher Scientific (Hampton, NH, USA). Ultra-pure water (18.2 MΩ·cm) was generated using a Milli-Q^®^ Integral system (MilliporeSigma, Burlington, MA, USA). Sample preparation involved a refrigerated centrifuge (Eppendorf 5430 R, Hamburg, Germany), ultrasonic processor (KQ5200E, Kunshan Ultrasonic Instruments Co., Ltd., Kunshan, China), and a Thermo Scientific™ Orbitrap ID-X Tribrid™ mass spectrometer equipped with a heated electrospray ionization (HESI) source (Thermo Fisher Scientific, San Jose, CA, USA).

#### 2.6.2. Sample Preparation

Serum aliquots (100 μL) were subjected to a methanol-based extraction protocol: (1) rapid thawing at 4 °C; (2) protein precipitation with 900 μL ice-cold methanol (1:9 *v/v*) with 1 min of vortex mixing; (3) ultrasonication in an ice bath (20 min); (4) centrifugation (13,000× *g*, 4 °C, 10 min); and (5) secondary centrifugation (13,000× *g*, 4 °C, 5 min) for supernatant clarification. Dried residues were reconstituted in 300 μL methanol (0.1% formic acid) and filtered through 0.22 μm PTFE membranes (MilliporeSigma, MA, USA). Pooled quality control (QC) samples were generated by combining 30 μL aliquots from each specimen.

#### 2.6.3. Chromatographic Conditions

Chromatographic separation utilized an Acquity UPLC BEH C18 column (100 mm × 2.1 mm, 1.8 μm; Waters, Milford, MA, USA) maintained at 35 °C. Mobile phases consisted of (A) 0.1% formic acid in water and (B) 0.1% formic acid in acetonitrile. The following gradient elution program was implemented: 5–20% B (0–2.0 min), 20–60% B (2.0–5.0 min), 60–99% B (5.0–6.0 min), 99% B (6.0–7.5 min), returning to 5% B (7.6–10.0 min) at 0.4 mL/min flow rate. The injection volume was 5 μL.

#### 2.6.4. Mass Spectrometric Parameters

Data acquisition employed dual-polarity HESI mode with the following settings:

Positive ion mode: spray voltage 3.8 kV, capillary temperature 325 °C, vaporizer temperature 400 °C, sheath gas 50 arb, and auxiliary gas 10 arb; full-scan MS (*m*/*z* 100–1000) at 120,000 resolution; HCD fragmentation at normalized collision energies (15%, 30%, 45%) with 8 dependent scans; and negative ion mode: spray voltage 3.0 kV with identical thermal parameters. Mass axis calibration used leucine enkephalin ([M+H]^+^: 556.2771; [M−H]^−^: 554.2627). Preanalysis mass calibration employed FlexMix™ solution (Thermo Scientific).

#### 2.6.5. Quality Assurance Protocol

System suitability testing involved (1) 10 blank (methanol) injections and (2) 10 consecutive QC injections for column equilibration. Experimental samples were analyzed in randomized order with QC injections every 10 samples. Data acquisition required QC peak intensity RSD < 15% and retention time drift < 0.1 min.

### 2.7. Gut Microbiota 16S rRNA Detection

The 16S rRNA gene sequencing analysis was conducted in collaboration with BGI Genomics (Shenzhen, China). Fecal DNA extraction was performed using the DNeasy PowerSoil Pro Kit (Qiagen Inc., Germantown, MD, USA), following the manufacturer’s protocol with bead-beating homogenization (5 min at 30 Hz) to ensure microbial lysis efficiency. DNA purity and concentration were quantified using a NanoDrop 2000 spectrophotometer (Thermo Fisher Scientific Inc., Waltham, MA, USA), with A260/A280 ratios maintained between 1.8 and 2.0. DNA integrity was verified with 1% agarose gel electrophoresis (100 V, 40 min), demonstrating distinct bands > 15 kb.

A total of 30 ng of qualified genomic DNA samples and the corresponding fusion primers were taken to configure the PCR reaction system, and the PCR reaction parameters were set for PCR expansion. The PCR amplification products were purified using Agencourt AMPure XP magnetic beads (Beckman Coulter, Inc., Brea, CA, USA) and dissolved in elution buffer and labeled, completing the library construction. The fragment range and concentration of the library were detected using an Agilent 2100 Bioanalyzer (Agilent Technologies, Inc., Santa Clara, CA, USA). Qualified libraries are selected based on the size of the inserted fragments. Sequencing was performed on the HiSeq platform (Illumina, Inc., San Diego, CA, USA).

### 2.8. Data Analysis

#### 2.8.1. Omics Data Preprocessing

Raw mass spectrometry data were processed using Progenesis QI software (v3.0, Waters Corporation, MA, USA) with a standardized workflow: (1) peak detection with adaptive sensitivity thresholds (minimum peak width = 0.1 min); (2) retention time alignment using a dynamic time warping algorithm; (3) baseline correction using asymmetric least squares smoothing; (4) spectral deconvolution with ApexTrack™ integration; and (5) intra-batch normalization based on total ion current. The three-dimensional matrix (*m*/*z*-retention time-peak intensity) was annotated against the Human Metabolome Database (HMDB, https://hmdb.ca/) (accessed on 15 September 2024) and LIPID MAPS^®^ database (https://www.lipidmaps.org) (accessed on 20 October 2024) using the following dual-filter criteria: (i) mass accuracy tolerance ≤ 5 ppm and (ii) isotopic pattern similarity score >80%. Nonendogenous metabolites (pharmaceuticals, environmental xenobiotics, and dietary biomarkers) were systematically excluded with database cross-referencing and manual validation.

The raw sequencing data were processed as follows to obtain clean data: (1) for the reads that match the primers, the software cutadapt v2.6 was used to remove the primers and connectors to obtain the fragments of the target area; (2) the method of removing low-quality reads was performed, with the window length set as 30 bp. If the average quality value of the window is less than 20, the read end sequence starting from the window is removed, and reads with a final read length less than 75% of the original read length are excluded; (3) reads containing N were removed; (4) low-complexity reads (10 consecutive ATCGs) were removed to obtain the final clean data. The software USEARCH (v7.0.1090) was used to cluster the spliced Tags into OTUs. The main process is as follows: (1) clustering using UPARSE at a 97% similarity to obtain the representative OTU sequences; (2) using UCHIME (v4.2.40) to remove the chimeras generated by PCR amplification from the representative OTU sequences; and (3) using the usearch_global method to compare all tags with the OTU representative sequence and obtain the abundance statistics on OTUs for each sample.

#### 2.8.2. Statistical Analysis

Continuous variables with a normal distribution were expressed as means ± standard deviations (SDs), whereas non-normally distributed data were reported as medians with interquartile ranges (IQRs). Categorical variables were summarized as frequencies with percentages (*n* (%)). For intergroup comparisons, independent two-group analyses of normally distributed data utilized Student’s *t*-test with Levene’s test for homogeneity of variance. Multi-group comparisons (≥3 groups) employed one-way analysis of variance (ANOVA) with post hoc LSD tests for normally distributed data. Kruskal–Wallis test with Dunn’s correction (≥3 groups) was used for skewed distributions. Categorical variable comparisons were performed using Pearson’s chi-square test or Fisher’s exact test for small expected frequencies (<5). All statistical analyses were performed using SAS software (v.9.4; SAS Institute Inc.; Cary, NC, USA).

To explore the relationship between childhood obesity and metabolites, multivariate logistic regression analysis was performed using the OVOB group or the MUO group as the dependent variable. Independent variables included demographic characteristics, early nutritional factors, dietary behaviors with statistically significant differences in the univariate analysis, and differential metabolites obtained from the OPLS-DA analysis. For the independent variables identified in the multivariate analysis (*p* < 0.05), the random forest (RF) model was used to predict the importance of differential metabolites related to OVOB or MUO. In this study, the differential metabolites of the OPLS-DA analysis were taken as independent variables, and OVOB or MUO were taken as dependent variables. The importance of the variables was quantified based on the mean decrease in the Gini index, and the RF model was trained with 500 trees, and feature importance was validated using 10-fold cross-validation repeated 5 times to ensure stability. Permutation tests (*n* = 1000) were performed to assess significance of AUC values. The RF model was implemented using the “random forest” package (version 4.7-1.1) in R version 4.5.0 (R Foundation for Statistical Computing, Vienna, Austria). A graphical receiver operating characteristic (ROC) curve was generated, and the area under the ROC curve (AUC) was utilized as a performance metric to assess the model’s effectiveness. A *p* value of less than 0.05, based on two-tailed test results, was considered statistically significant. All analyses were conducted using R version 4.5.0.

Metabolomic analysis was performed using MetaboAnalyst 6.0 (https://www.metaboanalyst.ca/) (accessed on 15 November 2024) to conduct multivariate statistical analysis and data visualization of sample differences. Principal component analysis (PCA) and partial least squares discriminant analysis (PLS-DA) were initially employed to characterize metabolic profile variations. Subsequently, orthogonal projections to latent structures discriminant analysis (OPLS-DA) was implemented for rigorous screening of differential metabolites between experimental groups. The following selection criteria were applied: variable importance in projection (VIP) score ≥ 1, false discovery rate (FDR) < 0.05, and significant fold change (FC) thresholds. Metabolic pathway enrichment analysis was subsequently conducted on the identified differential metabolites.

For gut microbiota characterization, α and β diversity analyses were performed using Prism 9.0 (GraphPad Software, Inc., San Diego, CA, USA). The α-diversity was assessed using four complementary indices: Ace (abundance-based coverage estimator), Chao1 (richness estimator), Shannon (diversity index), and Simpson (dominance index). Meanwhile, β diversity was determined using analysis of similarities (ANOSIM) based on Bray–Curtis distance matrices. We removed taxa existing in <10% of the samples to avoid the zero-inflated distributions, reducing the dataset from 389 to 209 genera (Appendix A). Owing to the inherent sparsity and limited sample size of the data in our exploratory study, we used MaAsLin2 (version 1.22.0; Huttenhower Lab, Harvard T.H. Chan School of Public Health, Boston, MA, USA) to analyze the association between differential abundance and study groups. MaAsLin2 analysis was performed using total sum scaling (TSS) normalization with Log2 transformation to account for compositional data characteristics. The analysis method was performed using a linear model (LM). Covariates (fixed_effects: age, sex, and dietary) were included in the model to control for confounding effects, and multiple test corrections were performed using the Benjamini–Hochberg method (BH); associations with FDR < 0.10 were considered exploratory given the small sample size and should be interpreted with caution. Additionally, we have implemented random effects in our computational approach to more accurately capture between-individual and between-group variations, thereby enhancing the reliability of our results. The normalized microbial data were imported into R (version 4.5.0). Visualization was performed using ggplot2 (version 3.5.2). Finally, Spearman’s rank correlation analysis (*p* < 0.05) was conducted to elucidate potential interactions between differential metabolites and discriminant bacterial taxa.

## 3. Results

### 3.1. Characteristics of Study Participants

The study comprised 285 children aged 5–7 years (73.2 ± 5.8 month; 164 boys (57.5%)), stratified into five BMIZ and metabolic phenotype-matched groups: WAS (*n* = 55), MHWH (*n* = 54), MUWH (*n* = 67), MHO (*n* = 36), and MUO (*n* = 73). Demographic characteristics revealed comparable age and gender distributions across groups (*p* = 0.149 and *p* = 0.868, respectively). However, significant intergroup variations emerged in geographic distribution (Henan/Guizhou), birth weight, breastfeeding ratio, and breastfeeding duration (all *p* < 0.05; Table 1).

Birth weight demonstrated significant associations with subsequent weight status independent of metabolic health. The OVOB subgroup exhibited elevated birth weights compared to the NOR and WAS groups (3.43 ± 0.41 kg vs. 3.26 ± 0.42 kg for NOR and 3.08 ± 0.47 kg for WAS; *p* < 0.001). Similar patterns emerged when comparing metabolic phenotypes. MHO participants had higher birth weights than MHWH counterparts (3.54 ± 0.43 kg vs. 3.21 ± 0.39 kg, *p* = 0.019), whereas MUO subjects showed greater birth weights than MUWH peers (3.49 ± 0.41 kg vs. 3.32 ± 0.38 kg, *p* = 0.046). Notably, metabolic health status did not exhibit these associations (MHWH vs. MUWH: *p* = 0.845; MHO vs. MUO: *p* = 0.536). Breastfeeding patterns revealed striking disparities, particularly in the MUO group, which demonstrated substantially higher non-breastfeeding rates (35.4%) compared to other groups (6.5–12.1%). Short-term breastfeeding (≤6 months) predominated in MUO participants (33.9%) relative to the MHO (19.4%) and MUWH (16.7%) groups (Table 1). These findings suggest potential dual impacts of breastfeeding duration on both weight trajectories and metabolic health development. No significant between-group differences emerged in perinatal factors (preterm birth, delivery mode), family structure, or socioeconomic indicators (caregiver education, occupation, and household income; Table A1).

Serum biochemical profiles across metabolic phenotype groups are summarized in Table A2. The MUO group exhibited significantly elevated TG levels (1.67 ± 1.12 mmol/L) compared to other groups (overall *p* < 0.001). LDL-C levels differed significantly among groups (*p* = 0.03), with MHO individuals showing higher values (1.80 ± 0.37 mmol/L) than MUWH participants (1.50 ± 0.53 mmol/L; *p* = 0.003). FBG levels varied significantly across phenotypes (*p* = 0.03). Both MUO (5.61 ± 1.48 mmol/L) and MUWH (5.58 ± 1.71 mmol/L) groups had higher glucose concentrations than their metabolically healthy counterparts (MHO: 5.02 ± 0.93 mmol/L; MHWH: 5.06 ± 0.80 mmol/L; MUO vs. MHO: *p* = 0.035; MUWH vs. MHWH: *p* = 0.036); however, no significant differences were observed between metabolically concordant weight groups (MHO vs. MHWH: *p* = 0.910; MUO vs. MUWH: *p* = 0.903). Metabolic abnormality groups (MUO: 116.83 ± 9.87 mmHg; MUWH: 118.15 ± 8.47 mmHg) exhibited significantly higher SBP than metabolically normal groups (MHO: 107.47 ± 5.12 mmHg; MHWH: 109.20 ± 6.80 mmHg; all *p* < 0.01). Metabolically abnormal groups (MUO: 77.83 ± 8.21 mmHg; MUWH: 78.68 ± 7.76 mmHg) showed elevated DBP compared to metabolically normal groups (MHO: 67.47 ± 5.12 mmHg; MHWH: 69.20 ± 6.80 mmHg; all *p* < 0.01). No differences were observed between metabolically concordant weight groups (MHO vs. MHWH: SBP *p* = 0.325, DBP *p* = 0.274; MUO vs. MUWH: SBP *p* = 0.546, DBP *p* = 0.502). Furthermore, we also observed differences in biochemical outcomes related to obesity such as ADP, ALT, and CRP-H, as detailed in Table A2.

### 3.2. Metabolomic Characteristics

#### 3.2.1. Metabolic Profile Characteristics

Initial metabolic profiling using principal component analysis (PCA) demonstrated robust analytical reliability. Tight clustering of quality control (QC) samples is observed (Figure A1), confirming system stability throughout the experimental workflow. Although preliminary PCA showed marginal intergroup separation in comparative analyses (OVOB vs. NOR, OVOB vs. WAS, and MUO vs, MHO), subsequent orthogonal partial least squares discriminant analysis (OPLS-DA) achieved effective metabolic profile discrimination across all paired groups: OVOB/NOR (R^2^Y = 0.81, Q^2^ = 0.69), OVOB/WAS (R^2^Y = 0.78, Q^2^ = 0.65), and MUO/MHO (R^2^Y = 0.73, Q^2^ = 0.61). Model validity was rigorously confirmed using permutation testing (200 iterations, *p* < 0.001), demonstrating resistance to overfitting (Figure 2 and Appendix A). These statistically robust OPLS-DA models establish a credible foundation for identifying biologically relevant differential metabolites between comparison groups.

#### 3.2.2. Classification of Differential Metabolites

Applying stringent bioinformatic thresholds (VIP ≥ 1.0, FDR < 0.05, and FC > 1.2/<0.83), we systematically identified 225 differential metabolites across comparative groups. The predominant metabolite classes included glycerophospholipids (44 metabolites, 19.4%), carboxylic acids and derivatives (43 metabolites, 18.9%), fatty acyls (33 metabolites, 14.5%), steroids and steroid derivatives (18 metabolites, 7.9%), and prenol lipids (16 metabolites, 7.0%). Comparative analysis revealed substantial intergroup heterogeneity in metabolite class distribution (Figure 3). The composition of differential metabolites in the OVOB/NOR group and the OVOB/WAS group is basically consistent with the top five category compositions of all differential metabolites, whereas the composition of differential metabolites in the MUO/MHO group is significantly different from that of other groups. Steroids and their derivatives (MUO/MHO: 2.12%; OVOB/NOR and OVOB/WAS: 7.9–14.1%) and terpene lipids (4.25% vs. 7.0–9.6%) were significantly lower in the MUO/MHO group compared with other groups.

#### 3.2.3. Screening of Differential Metabolites and Pathway Enrichment Analysis

Heatmap visualization of metabolites surpassing stringent fold-change thresholds (FC > 1.5/<0.67) unveiled three distinct obesity-associated metabolic signatures across comparative groups (Figure 4). We observed that lysophosphatidylcholines, such as LysoPC (18:4) (FC = 2.23) and LysoPC (14:1) (FC = 2.16), and phosphatidylserines, such as PS (17:0/0:0) (FC = 1.72) and PS (16:0/16:0) (FC = 1.55), were upregulated in the OVOB group. Inositol 1,3,4-trisphosphate (FC = 0.58), inositol 1,4,5-trisphosphate (FC = 0.64), PE (24:0/14:0) (FC = 0.59), and PE (16:0/16:0) (FC = 0.39) were significantly downregulated in the OVOB group (FDR < 0.05). We also observed a similar situation based on the heatmaps. PS (17:0/0:0) (FC = 2.07), PS (16:0/16:0) (FC = 1.91), and LysoPC (20:5) (FC = 1.65) were enriched in the OVOB group, whereas PE (16:0/16:0) (FC = 51) and 1,3,4-trisphosphate (FC = 0.51) were enriched in the WAS group. Notably, different findings were observed in the MUO/MHO group. LysoPC (O-18:0/0:0) (FC = 1.45), LysoPC (16:1) (FC = 1.24), C16 sphinganine (FC = 1.27), sphinganine (FC = 1.24), phytosphingosine (FC = 1.24), neurine (FC = 1.24), and aminoadipic acid (FC = 1.28) were enriched in the MUO group. PE (16:0/20:4) (FC = 0.78), PI (P-16:0/15:0) (FC = 0.82), and PI (16:0/14:1) (FC = 0.81) were downregulated in the MUO group. Furthermore, we observed that metabolites such as 11-cis-retinol, retinyl beta-glucuronide, and vitamin A were also upregulated in obese children.

As noted in Figure 5, the differential metabolites in the OVOB/NOR and OVOB/WAS comparisons were co-enriched in glycerophospholipid metabolism and inositol phosphate metabolism. In addition to being enriched in glycerophospholipid metabolism, the differential metabolites of the MUO/MHO comparison also specifically activated ether lipid metabolism (*p* = 0.0065), sphingolipid metabolism pathways (*p* = 0.0163), and lysine degradation (pathway impact = 0.11), as detailed in Figure 5. In the MUO/MHO comparison, we found that, in addition to LysoPC (16:1/0:0), which was also identified in the glycerol phospholipid metabolic pathway in previous group comparisons, neurine and PE(16:0/20:4) were also enriched. Key differential metabolites involved in ether lipid metabolism included LysoPC (O-18:0/0:0), while phytosphingosine and sphinganine were enriched in the sphingolipid metabolism pathway. Additionally, aminoadipic acid was identified as the key differential metabolite of lysine degradation (Figure 5).

#### 3.2.4. Prediction of the Clinical Efficacy of Differential Metabolites

Integrated multivariate logistic regression models incorporating demographic characteristics, early-life nutritional factors, dietary behaviors, and physical activity patterns (Table A2 and Table A3) identified statistically significant variables (*p* < 0.05). These variables were subsequently evaluated using random forest modeling to determine variable importance (Figure 6 and Figure 7). The random forest algorithm quantified variable importance based on the mean decrease in the Gini index.

For OVOB/NOR (Figure 6), retinyl β-glucuronide, coroloside, 17:0 cholesteryl ester, cyclocalopin E, LysoPC(14:1), cortisone acetate, vitamin A, body weight, CRP-H, and sulfolithocholyl glycine were identified as top predictors. Diagnostic performance of selected biomarkers was assessed using receiver operating characteristic (ROC) curve analysis. ALT, CRP_H, and the identified metabolites predicted OVOB in children with an AUC of 0.888 (95% CI: 0.847, 0.928), a sensitivity of 76.0%, and a specificity of 86.2%, but external study is needed to confirm generalizability.

For MUO/MHO (Figure 7), PI (16:0/14:1), treosulfan, N’-formylkynurenine, LysoPE (16:0/0:0), elenaic acid, hexaethylene glycol, PS (P-20:0/0:0), ALT, and CRP_H were identified as critical discriminators. These potential biomarkers predicted MUO in obese children with an AUC of 0.967 (95% CI: 0.936, 0.998), a sensitivity of 88.9%, and a specificity of 95.9%. The high AUC (0.967) in our internal validation suggests potential discriminative power, but external studies are needed to confirm generalizability.

### 3.3. Gut Microbiome Characteristics

#### 3.3.1. Structural Characteristics of the Gut Microbiome

The gut microbiome analysis was performed for 42 children (median age: 76.6 months; 59.5% boys), with no significant differences in age, sex, delivery method, whether breastfeeding, and breastfeeding duration across WAS, NOR, and OVOB groups (*p* > 0.05). The diet survey showed that the energy intake of participants was 1320.4 (976.2,1637.5) kcal. Energy intake, grains and tubers, vegetables, fruits, meat, milk and dairy products, eggs, fish, snacks high in sugar and fat, and sugary drinks did not significantly differ among the WAS, NOR, and OVOB groups (*p* > 0.05). For details, please refer to Section A.4 (Table A4). In addition, 16S rRNA analysis showed that dominant phyla comprised Firmicutes (43.1%), Bacteroidetes (40.1%), and Actinobacteria (11.2%), with *Bacteroides* (24.4%) and *Prevotella* (14.4%) exhibiting genus-level dominance (Figure 8).

Intergroup comparative analysis demonstrated two key structural divergences (Figure 9). The Firmicutes/Bacteroidetes ratio was elevated (2.07 in OVOB vs. 1.24 in NOR and 0.94 in WAS; F = 7.86, *p* = 0.009). In addition, Verrucomicrobia was depleted (0.04% in OVOB vs. 0.29% in NOR and 0.01% in WAS; *p* = 0.0053). The Verrucomicrobia abundance ranged from high to low in the following order: NOR (0.29%), OVOB (0.04%), and WAS group (0.01%). Pairwise comparisons showed statistically significant differences (*p* = 0.005).

#### 3.3.2. Microbial Diversity Signatures

Comparative analysis revealed distinct α/β diversity patterns across different groups, and statistical differences in α/β diversity patterns were noted among the three groups (Figure 10). Pairwise comparison showed that, compared to NOR controls, obese children exhibited species richness reduction (Ace index: 395.25 ± 85.25 vs. 502.34 ± 92.17 (*p* = 0.0007), Chao1 index: 379.51 ± 75.78 vs. 490.39 ± 78.19 (*p* = 0.0013)) and a decline in community evenness (Shannon index: 2.96 ± 0.57 vs. 3.45 ± 0.62 (*p* = 0.03) and Simpson index: 0.12 ± 0.04 vs. 0.07 ± 0.05 (*p* = 0.01)). β diversity showed significant structural divergence (ANOSIM R = 0.114, *p* = 0.047). In addition, the Ace and Chao1 indices of children in the WAS group were lower than those in the NOR group (*p* values were < 0.0001 and 0.001, respectively); however, no statistically significant difference in the Shannon and Simpson indices (*p* > 0.05) were observed, indicating that the abundance of microbiota in emaciated children was lower than that of normal children (Figure 10).

#### 3.3.3. Differential Bacteria Screening

MaAsLin 2 analysis was performed to identify differentially abundant genera across groups. In the association analysis of the OVOB, WAS, and NOR groups, a total of 12 differential bacterial genera were identified (FDR < 0.10). As shown in Figure 11, compared with the NOR group, the OVOB group exhibited significant correlations with bacterial genera, including negative correlations with SCFA-producing bacterial genera, such as *Colidextribacter*, *Dysosmobacter*, *Intestinimonas*, *Longicatena*, and *Butyricimonas* (coefficients of −3.20, −3.03, −2.98, −1.04, and −2.92, respectively; FDR values of 0.02, 0.02, 0.02, 0.08, and 0.09, respectively). *Alistipes* were negatively correlated with children in the OVOB group (coefficients = −2.28 and FDR = 0.06). Associations with FDR < 0.10 were considered exploratory given the small sample size and should be interpreted with caution.

### 3.4. Correlation Analysis Between Differential Bacteria and Differential Metabolites

In the OVOB group, significant correlations were observed between differential metabolites and microbial genera (Figure 12). PE (16:0/16:0) demonstrated positive correlations with multiple genera, including *Butyricimonas* (r = 0.448, *p* = 0.008), *Dysosmobacter* (r = 0.483, *p* = 0.003), *Intestinimonas* (r = 0.411, *p* = 0.015), *Longicatena* (r = 0.386, *p* = 0.024), and *Alistipes* (r = 0.422, *p* = 0.012). Methyl sorbate was positively associated with *Butyricimonas* (r = 0.432, *p* = 0.011) and *Intestinimonas* (r = 0.413, *p* = 0.015), whereas deacetylnomilin displayed negative correlations with *Intestinimonas* (r = −0.349, *p* = 0.043) and *Alistipes* (r = −0.399, *p* = 0.019).

## 4. Discussion

This multi-omics study explored the metabolic and gut microbiota characteristics associated with the weight and metabolic phenotypes of obese children and analyzed the associations between different metabolites and different bacterial genera. These findings provide a new perspective for understanding the heterogeneity in childhood obesity. However, it is necessary to note the limitations of the study population. Our study included children from four districts and counties in central and western China. The population characteristics were relatively simple (such as similar diet, cultural background, and environmental exposure). Although this improved the comparability of the results, it also limited the universality of the conclusions. The correlation between the gut microbiome and obesity is also influenced by multiple factors such as race, age, region, diet, and environmental exposure [24]. Yan He found that the influence of regional factors on the microbiota was significantly greater than that of other factors such as age, disease, and lifestyle [34]. The influence of different geographical locations on the composition of gut microbiota may stem from differences in lifestyle and dietary culture. For instance, a “geographical gradient” is noted among the gut microbiota of European infants, which is manifested in the higher abundance of Bifidobacterium and Clostridium in infants from Northern Europe (such as Denmark, Sweden, and the United Kingdom). In Southern Europe (such as Spain and Portugal), the abundance of Lactobacillus and Bacteroides in the intestines of infants is relatively high [35]. Meanwhile, the dietary patterns of different ethnic groups and different regions vary greatly, and dietary patterns affect the structural characteristics of the gut microbiota [26]; these differences have a significant impact on the association between the gut microbiota and obesity [18,24,36]. Therefore, the results of this study need to be further verified in populations representing other races and regions.

Our findings reaffirm the fetal origins of obesity [10], with elevated birth weight independently predicting adiposity (OVOB: 3.43 vs. NOR: 3.26 kg, *p* < 0.001), aligning with the fetal overnutrition hypothesis. Breastfeeding duration emerged as a dual modifier, reducing both obesity risk (MHO short-term breastfeeding: 19.4% vs. MUO: 33.9%) and metabolic dysregulation (MUO non-breastfeeding rate: 35.4% vs. <12% in other groups). The systematic review revealed that, compared with formula-fed or mixed-fed children, children who were exclusively breastfed from 3 to 12 months of age exhibited lower BMI trajectories or a reduced likelihood of following a high BMI trajectory. Furthermore, the results indicated that an extended duration of breastfeeding was associated with a progressively lower BMI trajectory until the age of 18 [37].

We found that the abundance of steroid derivatives and terpenoids in children with MUO decreased. A study involving 191 obese children aged 5 to 18 show that childhood obesity, especially prepubertal obesity, is associated with several steroid changes caused by excessive weight [15]. They defined childhood obesity as a “steroid-related complex disease”, and the metabolomics characteristics of urinary steroids in obese children revealed the targeted characteristics of 31 steroid metabolites [38]. Studies have shown that the lipid marker LysoPC promotes adipocyte proliferation through PPARγ activation in vitro [39], and the upregulation of LysoPC may be related to the downregulation of genes and proteins related to lipid transport and decomposition (PPARγ) [40]. This mechanism is consistent with the increase in LysoPC observed in obese children in our study. It is worth noting that downregulation of PI (16:0/14:1) in the MUO/MHO model has important significance. PI (16:0/14:1) is an anti-inflammatory phosphatidylinositol, and its depletion may impair the membrane anchoring ability of the insulin signaling pathway [41], which is directly related to the insulin resistance phenotype in children with MUO [42]. This molecule promotes the membrane localization of glucose transporter GLUT4 by binding to the PH domain of IRS-1, and its deletion may lead to impaired glucose uptake in skeletal muscle [43]. The high predictive power of the ROC curve (MUO AUC of 96.7%) suggests that a combination of metabolic markers (e.g., PI (16:0/14:1) and treosulfan) and liver function markers (ALT) can effectively distinguish MHO and MUO obesity. However, this research result still requires external verification.

The observed “low diversity-high Firmicutes/Bacteroidetes ratio” pattern in obese children mirrors obesity-associated dysbiosis [18]. As a bridge between obesity and intestinal homeostasis, SCFAs are closely related to the pathogenesis of obesity and related metabolic diseases [44]. Studies indicated that depletion of SCFA-producing genera in obese children may compromise intestinal barrier function and energy regulation [27], and we also found a reduced abundance of SCFA-producing genera in the OVOB group. We found a positive correlation between PE (16:0/16:0) and SCFA-producing genera, which is consistent with the finding that phosphatidylethanolamines enhance butyrate production in animal models [45]. However, cross-sectional data cannot determine whether microbial activity directly influences PE levels or whether shared dietary factors (e.g., fiber intake) independently affect both variables. Lysophospholipids, such as LysoPE, are known to modulate membrane permeability and inflammatory responses [46]; however, whether these correlations reflect microbial modulation of host lipid metabolism or vice versa remains unclear. Similarly, the negative correlation between deacetylnomilin and *Alistipes* parallels reports linking *Alistipes* to reduced secondary bile acid metabolism [47]. However, mechanistic studies are needed to clarify this relationship. Thus, our exploratory findings nominate multi-omics signatures for further validation as potential stratification tools, pending replication in larger cohorts.

## 5. Innovations and Limitations

This study innovatively combines metabolic phenotyping with multi-omics profiling in a pediatric cohort. The identification of terpenoid/steroid deficiencies and sphingolipid alterations in MUO children provides potential biomarkers for distinguishing high-risk subgroups.

However, this study also has limitations that should be noted: (1) the cross-sectional design precludes causal inferences; longitudinal cohorts are needed to track metabolite–microbiota dynamics during MUO progression. (2) The microbiome sample size (*n* = 42) may be underpowered to detect rare genera and increase overfitting risk; independent validation is essential. Subsequent studies can integrate multi-omics techniques (such as metagenomics, metabolomics, and epigenome) to further verify the regulatory effects of key metabolic pathways and functional targets of the microbiota.

## 6. Conclusions

This multi-omics study characterized the preliminary metabolic and gut microbial profiles of Chinese children aged 5–7 years with obesity and metabolic heterogeneity phenotypes. Our data suggest that MUO children may exhibit alterations in ether lipid metabolism and sphingolipid metabolism pathways, showed tentative associations with elevated LysoPCs, and reduced PEs/PIs. Obese children displayed reduced microbial diversity, an elevated Firmicutes/Bacteroidetes ratio, along with apparent reductions in certain SCFA-producing genera. Observed correlations between specific bacterial taxa and phospholipid metabolites could indicate microbial–host metabolic interactions, though alternative explanations, including shared dietary influences, cannot be excluded.

These exploratory results contribute to the growing understanding of metabolic heterogeneity in childhood obesity but should be interpreted with appropriate caution until replicated in more diverse populations and supported by functional validation.

## Figures and Tables

**Figure 1 nutrients-17-01876-f001:**
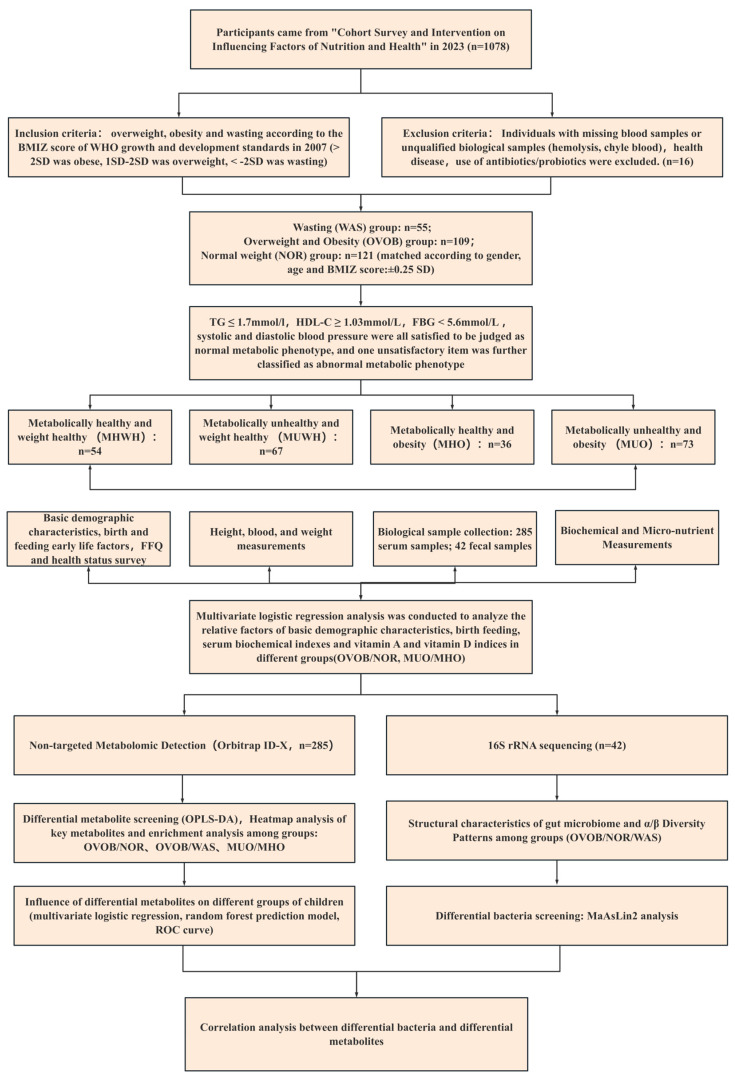
Research flow design chart.

**Figure 2 nutrients-17-01876-f002:**
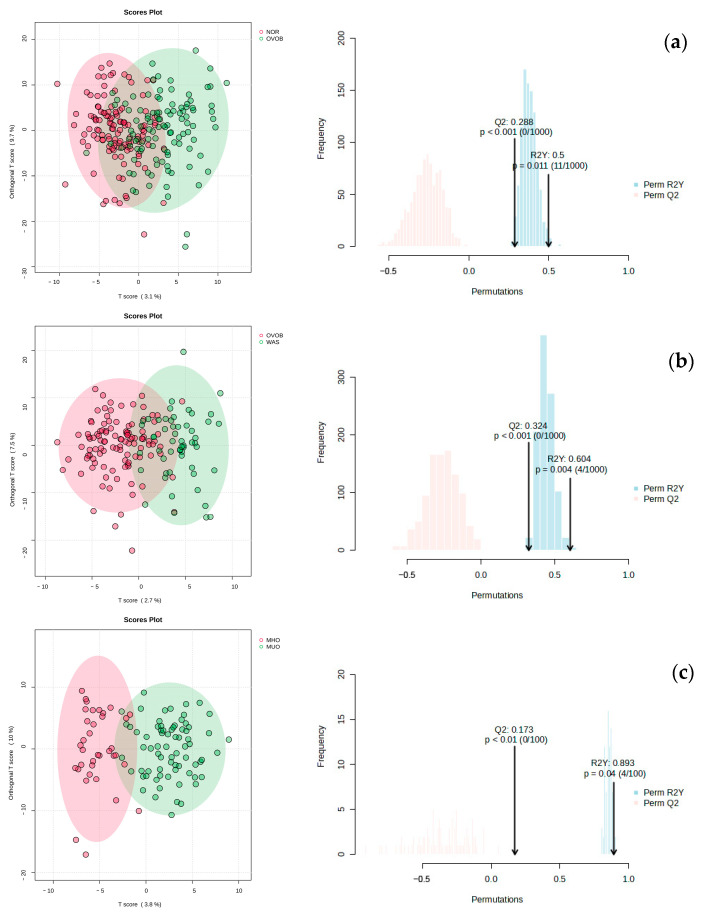
OPLS-DA model and permutation testing results of samples from comparison groups. Columns 1 and 2 are the results of metabolite group difference analysis using OPLS-DA and model permutation testing under positive ion modes, respectively. (**a**) Description of the results of the OPLS-DA and permutation tests for the OVOB/NOR comparison group; (**b**) description of the results of the OPLS-DA and permutation tests for the OVOB/WAS comparison group; (**c**) description of the results of the OPLS-DA and permutation tests for the MUO/MHO comparison group.

**Figure 3 nutrients-17-01876-f003:**
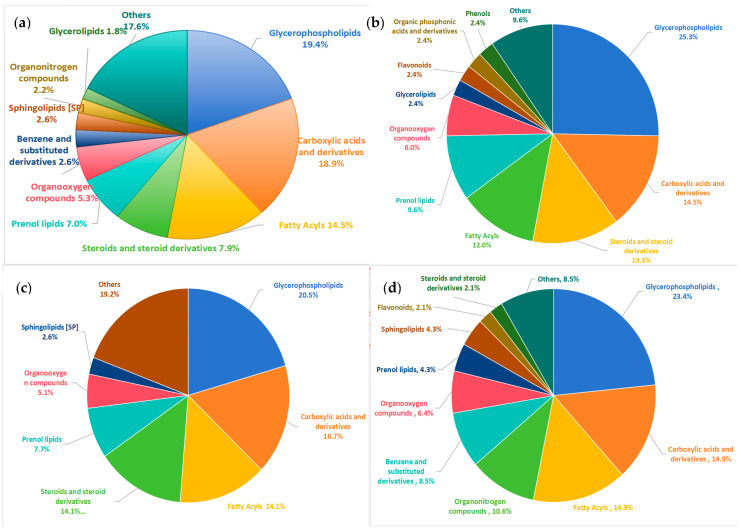
Class distribution of differential metabolites among comparative groups. Differential metabolites were analyzed using OPLS-DA and screened according to VIP ≥ 1, FDR < 0.05, FC > 1.2, or FC < 0.83. The primary classification of human metabolites was performed using The Human Metabolome Database (HMDB Version 5.0). (**a**) The distribution of differential metabolites in all groups; (**b**) the distribution of differential metabolites in the OVOB/NOR group; (**c**) the distribution of differential metabolites in the OVOB/WAS group; and (**d**) the distribution of differential metabolites in the MUO/MHO group.

**Figure 4 nutrients-17-01876-f004:**
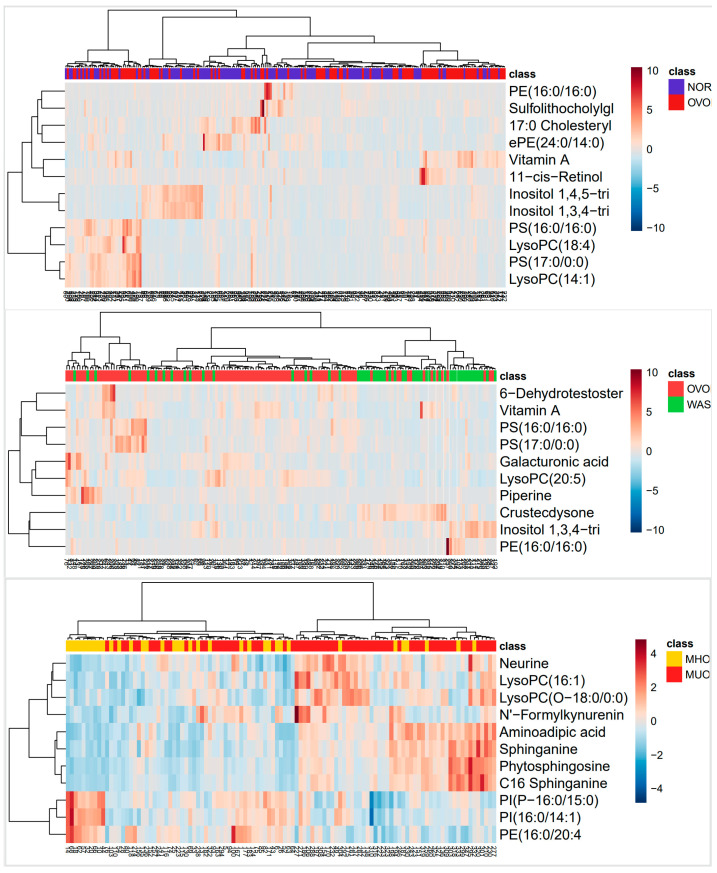
Heatmap analysis of different metabolites between comparative groups. Differential metabolites were screened for heatmap analysis across groups. Metabolites differentiating OVOB/NOR and OVOB/WAS were defined by an FC > 1.5 or FC < 0.67 (VIP ≥ 1, FDR < 0.05). Metabolites differentiating MUO/MHO were defined by an FC > 1.2 or FC < 0.83 (VIP ≥ 1, FDR < 0.05).

**Figure 5 nutrients-17-01876-f005:**
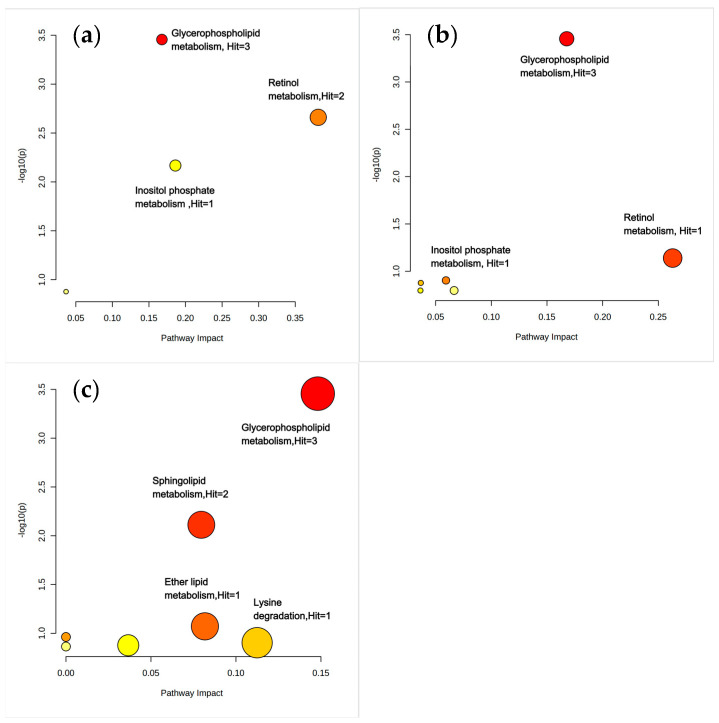
Pathway enrichment analysis of differential metabolites. Differential metabolites identified based on VIP ≥ 1, FDR < 0.05, and FC > 1.5/<0.67 were analyzed using MetaboAnalyst 6.0 to assess pathway enrichment. (**a**) Overview of metabolite enrichment sets and bubble diagram for OVOB/NOR, (**b**) OVOB/WAS, and (**c**) MUO/MHO.

**Figure 6 nutrients-17-01876-f006:**
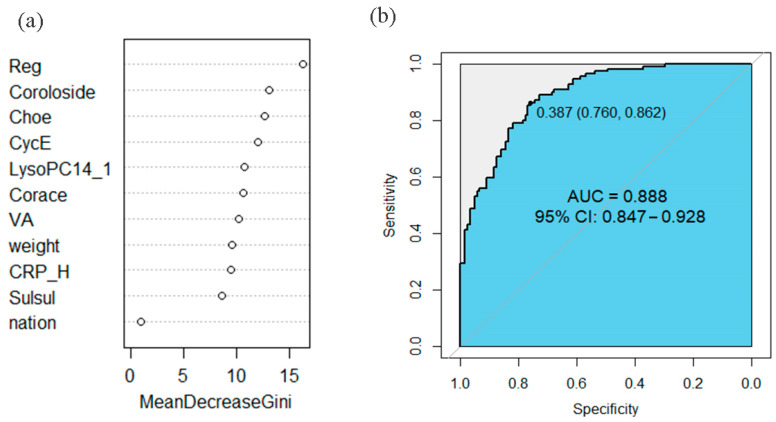
Importance ranking of differential metabolites and efficacy of predicting OVOB. (**a**) Random forest model for importance ranking of metabolites. The random forest algorithm quantified variable importance based on a mean decrease in the Gini index. (**b**) The combined efficacy of weight, nation, CRP_H, VA, and the identified metabolites in predicting OVOB in children had an AUC of 0.888 (95% CI: 0.847, 0.928), with a sensitivity of 76.0% and a specificity of 86.2%. CRP_H, C-reactive protein; Reg, retinyl beta-glucuronide; Choe, 17:0 cholesteryl ester; CycE, cyclocalopin E; LysoPC14_1, LysoPC (14:1); VA, vitamin A; Corace, cortisone acetate; Sulsul, sulfinpyrazone sulfide; weight, birth weight.

**Figure 7 nutrients-17-01876-f007:**
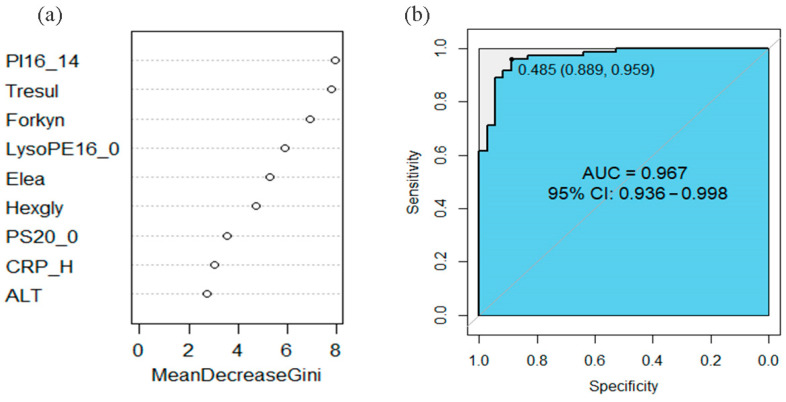
Importance ranking of differential metabolites and efficacy of predicting MUO. (**a**) Random forest model for importance ranking of metabolites. The random forest algorithm quantified variable importance based on a mean decrease in the Gini index. (**b**) The combined efficacy of ALT, CRP_H, and the metabolites in predicting MUO in obese children achieved an AUC of 0.967 (95% CI: 0.936, 0.998), with a sensitivity of 88.9% and a specificity of 95.9%. ALT, alanine aminotransferase; CRP_H, C-reactive protein; PI16_14, PI (16:0/14:1); Tresul, treosulfan; Forkyn, N′-formylkynurenine; LysoPE16_0, LysoPE (16:0/0:0); Elea, elenaic acid; Hexgly, hexaethylene glycol; PS20_0, PS (P-20:0/0:0).

**Figure 8 nutrients-17-01876-f008:**
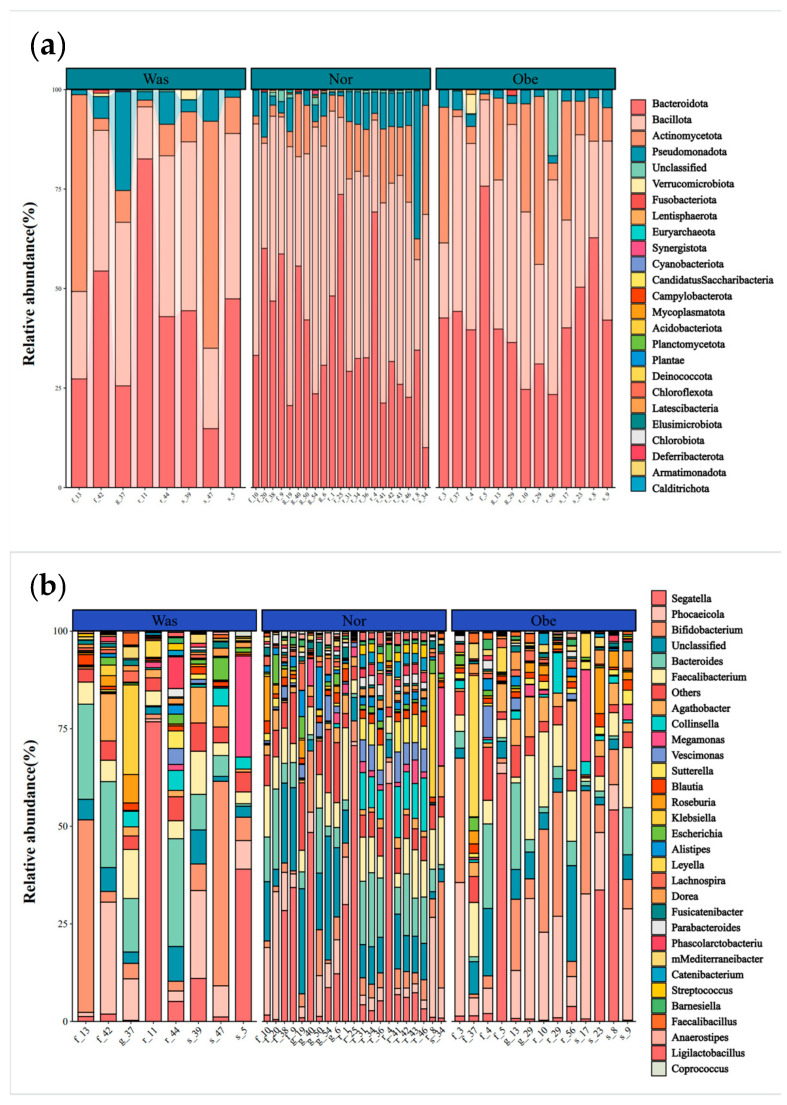
Relative abundance of gut microbiome in comparative groups. (**a**) Phylum level; (**b**) genus level.

**Figure 9 nutrients-17-01876-f009:**
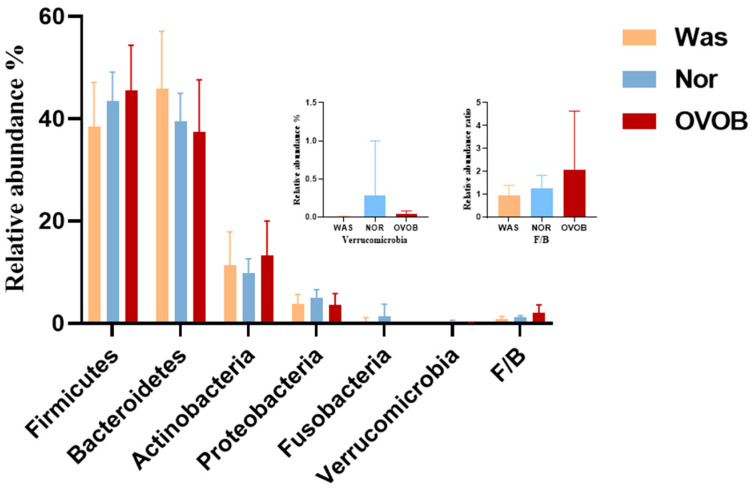
Analysis of relative abundance composition of gut microbiota in different groups. The Kruskal–Wallis test was used to analyze the difference in the F/B ratio and Verrucomicrobia abundance among the three groups.

**Figure 10 nutrients-17-01876-f010:**
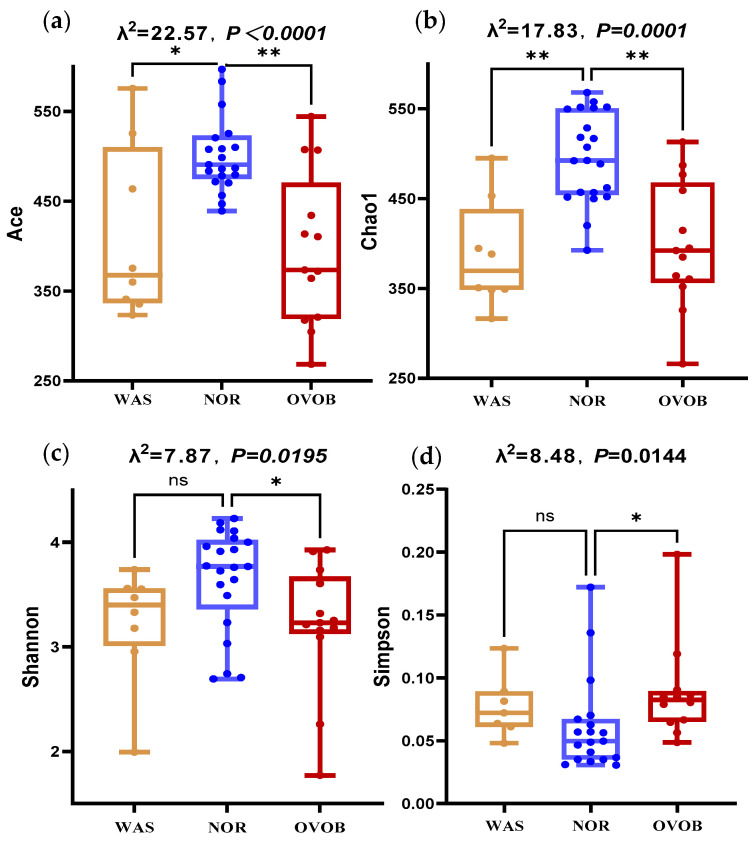
Box diagram of α diversity of gut microbiota in different groups. (**a**–**d**) represents the difference in the Ace, Chao1, Shannon, and Simpson indices among the three groups, respectively, and the difference was statistically significant among the three groups (*p* < 0.05). Kruskal–Wallis test was used to analyze the differences in the Ace, Chao1, Shannon, and Simpson indices. * indicates that the difference is statistically significant (*p* < 0.05), ** indicates that the difference is statistically significant (*p* < 0.01).

**Figure 11 nutrients-17-01876-f011:**
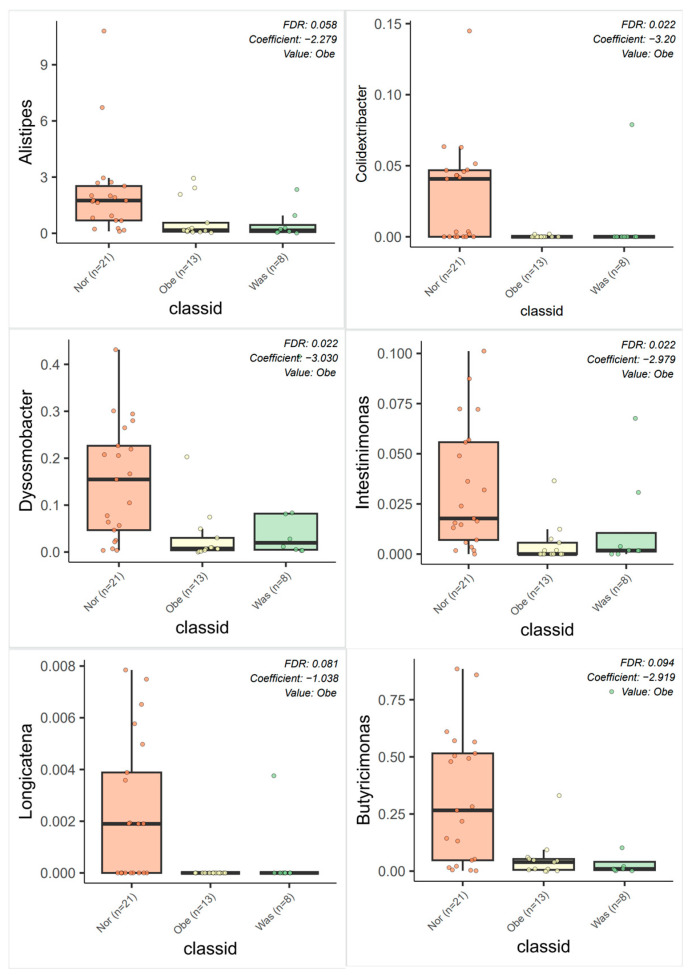
MaAsLin 2 analysis results of differential bacteria between comparative groups. The FDR of *Colidextribacter*, *Dysosmobacter*, and *Intestinimonas* was 0.02, 0.02, and 0.02, respectively; the FDR of *Alistipes, Longicatena*, and *Butyricimonas* was 0.06, 0.08, and 0.09, respectively. Associations with FDR < 0.10 were considered exploratory given the small sample size and should be interpreted with caution.

**Figure 12 nutrients-17-01876-f012:**
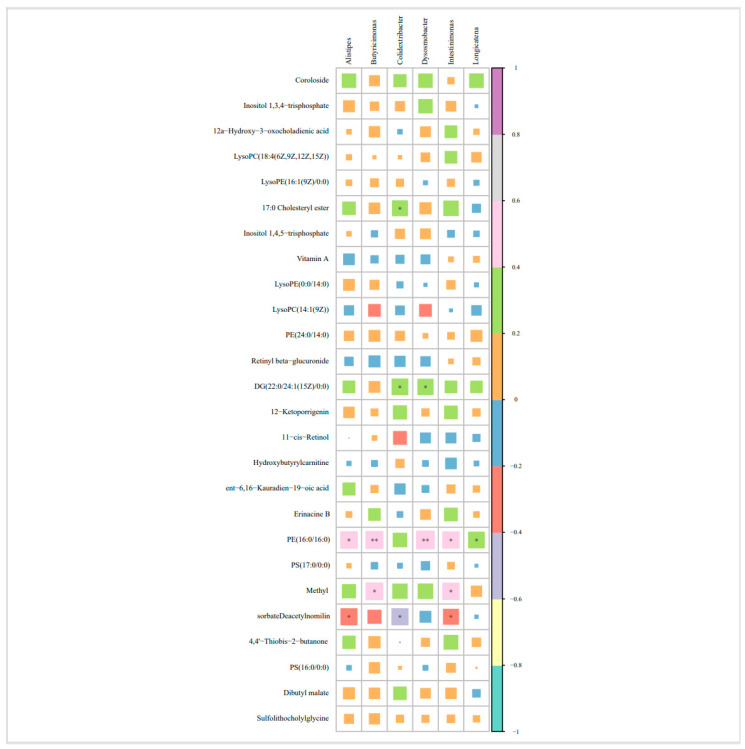
Spearman association between different metabolites and different bacteria in the OVOB/NOR group comparison. Square size represents the absolute value of the correlation coefficient (larger size = stronger association). * indicates that the correlation is statistically significant (*p* < 0.05), ** indicates that the correlation is statistically significant (*p* < 0.01).

**Table 1 nutrients-17-01876-t001:** Demographic characteristics and risk factors for obesity of the subjects.

	All (*n* = 285)	WAS (*n* = 55)	NOR			OVOB			F/χ^2^	*p*
			MHWH (*n* = 54)	MUWH (*n* = 67)	All (*n* = 121)	MHO (*n* = 36)	MUO (*n* = 73)	All (*n* = 109)		
Age ^a^, m	73.2 ± 5.8	72.6 ± 5.7	72.9 ± 4.7	73.2 ± 4.8	73.1 ± 4.8	73.9 ± 6.2	71.7 ± 6.6	72.4 ± 6.5	1.772	0.149
Gender ^b^										
Boy	164 (57.5)	30 (54.5)	30 (18.3)	38 (23.2)	68 (56.2)	20 (12.2)	46 (28)	66 (60.6)	1.261	0.868
Girl	121 (42.5)	25 (45.5)	24 (19.8)	29 (24)	53 (43.8)	16 (13.2)	27 (22.3)	43 (39.4)		
BMIZ ^a,c^	0.276 ± 1.86	−2.35 ± 0.39	−0.37 ± 0.65	−0.12 ± 0.68	−0.23 ± 0.68	2.1 ± 1.11	2.18 ± 1.06	2.16 ± 1.08	298.987	<0.001
District ^b^										
Henan	149 (100.0)	19 (12.8)	36 (24.2)	37 (24.8)	73 (49.0)	24 (16.1)	3 3(22.1)	57 (38.3)	16.098	0.003
Guizhou	136 (100.0)	36 (26.5)	18 (13.2)	30 (22.1)	48 (35.3)	12 (8.8)	40 (29.4)	52 (38.2)		
Nation ^b^										
Han Chinese	207 (100)	33 (15.9)	44 (21.3)	53 (25.6)	97 (59.1)	27 (13)	50 (24.2)	77 47.0)	8.685	0.069
Ethnic minorities	78 (100)	22 (28.2)	10 (12.8)	14 (17.9)	24 (19.8)	9 (11.5)	23 (29.5)	32 (26.4)		
Birth weight ^a^, kg	3.29 ± 0.44	3.08 ± 0.47	3.25 ± 0.40	3.26 ± 0.43	3.26 ± 0.42	3.47 ± 0.36	3.41 ± 0.43	3.43 ± 0.41	3.225	0.024
Birth height ^a^, cm	50.24 ± 1.26	49.83 ± 1.29	50.09 ± 1.21	50.38 ± 1.39	50.25 ± 1.32	50.48 ± 1.26	50.41 ± 1.06	50.44 ± 1.13	0.918	0.433
Breast feeding ^b^										
Yes	215 (86.0)	43 (93.5)	46 (90.2)	58 (87.9)	104 (88.9)	29 (93.5)	39 (69.6)	68 (78.2)	13.08	0.004
No	35 (14.0)	3 (6.5)	5 (9.8)	8 (12.1)	13 (11.1)	2 (6.5)	17 (30.4)	19 (21.8)		
Duration of breastfeeding ^b^, m										
≤6	54 (21.6)	7 (15.2)	11 (21.6)	11 (16.7)	22 (18.8)	6 (19.4)	19 (33.9)	25 (28.7)	35.06	<0.001
6–12	53 (21.2)	9 (19.6)	7 (13.7)	9 (13.6)	16 (13.7)	4 (12.9)	24 (42.9)	28 (32.2)		
≥12	143 (57.2)	30 (65.2)	33 (64.7)	46 (69.7)	79 (67.5)	21 (67.7)	13 (23.2)	34 (39.1)		

^a^ Normal distribution is presented as mean ± standard deviation. One-way analysis of variance (ANOVA) was used to test for differences between groups. ^b^ Categorical or count data are expressed as proportions/rates. Chi-square test was used to analyze differences between groups. ^c^ BMIZ: body mass index Z-scores.

## Data Availability

The original contributions presented in this study are included in the article. Further inquiries can be directed to the corresponding author.

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
