# Peer review of "Integrating Metabolomics and Gut Microbiota to Identify Key Biomarkers and Regulatory Pathways Underlying Metabolic Heterogeneity in Childhood Obesity"

_nutrients, 2025, doi:10.3390/nu17111876_

Round 1
Reviewer 1 Report
Comments and Suggestions for Authors
Thank you for the opportunity to review this manuscript entitled “Integrating Metabolomics and Gut Microbiota to Reveal Key Biomarkers and Regulatory Pathways of Metabolic Heterogeneity in Childhood Obesity.” This study addresses an important and timely topic by employing a multi-omics approach to dissect the metabolic and microbial underpinnings of pediatric obesity, with a specific focus on metabolic phenotypes. The integration of untargeted metabolomics and gut microbiota analysis is a strength of the study, and the large sample size is commendable. However, several critical issues related to methodological clarity, interpretative rigor, and organization need to be addressed to enhance the manuscript’s scientific robustness and overall readability. The major revisions below are intended to assist the authors in improving the quality and impact of their work.
-
The introduction provides useful context, but it lacks a precise formulation of the study hypothesis and objectives. The rationale for integrating metabolomic and microbiota data should be explicitly stated. The current version implies a hypothesis but does not clearly define it.
-
Several parts of the results (notably Sections 3.2.3, 3.2.4, and 4.2) contain overlapping descriptions of lipid and metabolite profiles. This redundancy hampers narrative flow. Consider streamlining these sections to present the findings more succinctly and avoid duplication.
-
The statistical approach involving random forest classification and ROC analysis lacks detail. It is unclear how variables were selected, whether feature importance was validated, and how overfitting was addressed. Cross-validation or external validation procedures should be reported. Additionally, provide metrics of model performance beyond AUC (e.g., sensitivity, specificity).
-
Certain statements in the discussion imply causal relationships between microbial taxa and metabolic pathways (e.g., SCFA-producing genera preventing lipid peroxidation), which cannot be inferred from cross-sectional data. These interpretations should be revised to clearly indicate that observed associations are correlational.
-
The criteria used to distinguish metabolically healthy vs. unhealthy obesity are limited to three metabolic markers. This definition diverges from more comprehensive standards in the literature and may compromise internal validity. Either justify the choice of markers more robustly or consider adopting an alternative classification system based on consensus definitions.
- Figures—especially heatmaps and ROC curves—are visually complex. Improve figure legends to more clearly explain the key takeaway of each visual. Where possible, simplify figure content or provide alternative visual summaries.
Reviewer 2 Report
Comments and Suggestions for Authors
The manuscript presents comprehensive and methodologically sound multi-omics research into the metabolic and microbial heterogeneity of childhood obesity. It provides important insights into phenotype-specific metabolic alterations and potential biomarkers.
Major Comments
- While the manuscript effectively identifies associations between specific metabolites/microbial taxa and obesity subtypes, the cross-sectional design limits causal inference. Statements like “ether lipid-sphingolipid axis as a pediatric-specific biomarker” and “pathways linking gut dysbiosis to lipid remodeling” may overstate the findings. Please revise the abstract, results, and discussion to use language such as “associated with” or “correlated with,” and clearly acknowledge the cross-sectional limitation.
- The manuscript refers to OTUs for clustering microbiota sequences, yet the actual pipeline (DADA2) denoises data into amplicon sequence variants (ASVs); a more precise and reproducible method. This must be corrected throughout the manuscript, including methods and results sections.
- The manuscript applies LEfSe to identify differentially abundant genera across groups. However, LEfSe has limitations in small sample sizes and high-sparsity datasets like microbiome data, especially with zero-inflated distributions. These limitations can lead to inflated false-positive rates. Given the modest cohort (n = 42 for microbiota analysis) and sparsity of data, LEfSe may not be statistically appropriate here. The authors are encouraged to consider more robust alternatives such as: MaAsLin2, which supports zero-inflated models and FDR correction. ANCOM or ANCOM-BC, which are specifically designed to handle microbiome compositionality and sparsity.
- While the manuscript uses rarefied data for alpha/beta diversity, rarefaction can discard valuable read information and reduce statistical power. For differential abundance analyses, rarefaction is generally discouraged in favor of methods that accommodate variable library sizes (e.g., cumulative sum scaling, DESeq2, or CLR transformation). Clarify how rarefaction was handled for each analysis type (diversity vs. differential abundance), and justify the approach taken, especially given the small sample size.
- The small number of fecal samples (n = 42) raises concerns regarding the stability of statistical inferences in microbiome analyses. Many genera likely have high sparsity or zero inflation, increasing the risk of spurious associations. The authors should provide the prevalence filtering threshold (e.g., ≥10% of samples), report the number of genera retained after filtering, clearly caveat their findings, especially those based on marginal p-values or rare taxa.
- Although age and sex are considered, additional covariates (e.g., dietary factors) might influence microbiota composition and should be evaluated or justified for exclusion.
- The discussion of lipidomic differences (e.g., sphingolipid accumulation in MUO vs. MHO) is compelling, but some mechanisms remain speculative. For example, links to TLR4/NF-κB signaling or mitochondrial peroxidation are not directly tested. Clarify when interpretations are hypothesis-generating and consider citing mechanistic studies that support the proposed roles of specific lipids and microbial interactions.
- The focus on a homogeneous Chinese cohort is a strength for internal validity but limits external generalizability. Dietary, genetic, and environmental factors vary across populations, and this should be more prominently addressed in the discussion.
- Some terms may confuse readers. For example, “ether lipid-sphingolipid axis” and “neuroactive metabolites” should be clearly defined on first mention. Use consistent metabolite names and abbreviations across figures, tables, and text. A supplementary table linking all metabolites and bacterial genera to their biological relevance would improve clarity.
- The conclusion mentions “precision diagnostics” and “phenotype-specific interventions.” While promising, these applications are premature without longitudinal or interventional data.
Reviewer 3 Report
Comments and Suggestions for Authors
First of all, I would like to emphasize the complexity with which the researchers approached the issue.
The second advantage is the subject chosen - childhood obesity is significant, because this disease is becoming a global problem and affects the health of the world's population.
I have a question: How were the children recruited for this study? The authors refer to the project, its number is provided - but it does not help to understand. I would like to know if these were screening tests or if these children went to their family doctor for a checkup. In order to be included in this group, the children had to be only 5-7 years old and obese, overweight, or wasting, but were any metabolic diseases ruled out? Were they treated before, were they put on a diet?)
How was the control group selected to be matched? Where were the healthy children chosen, and were they specially recruited for this study? Was their blood tested specifically for this purpose?
I suggest the authors add a section on the study's limitations in the article.
In Figure 1, the block marked MHWH (n=54) appears twice; this error should be corrected.
Fig. 3 - Some metabolite descriptions overlap on the pie chart - this should be corrected.
Fig. 8 - is illegible.
Fig. 10 - The description on the x-axis is completely illegible; it should be corrected and performed adequately
Other than that, I have no comments; the study is coherent, well-planned, and properly performed with due diligence.
The description of the methodology is correct, and the selection and performance of the statistical analyses do not raise any objections.
The presentation of the results, apart from the technical objections presented above, is correct. I have no objections regarding the discussion; the authors correctly comment on the obtained results and refer to the appropriate articles. The conclusions are correctly formulated. I do not have concerns regarding ethical aspects.
Research in this group is particularly difficult to conduct. Perhaps to increase the article's attractiveness, I suggest preparing an illustration with a diagram - something in the style of a graphical abstract.
Round 2
Reviewer 1 Report
Comments and Suggestions for Authors
The authors have addressed all my comments
Author Response
|
Response to Reviewer 1 Comments
|
||
|
1. Summary |
|
|
|
We sincerely appreciate the reviewer's time and valuable feedback, which has significantly improved our manuscript. We're grateful for their thoughtful comments and are pleased that our revisions have addressed all concerns.]
|
||
|
2. Questions for General Evaluation |
Reviewer’s Evaluation |
Response and Revisions |
|
Does the introduction provide sufficient background and include all relevant references? |
Yes/Can be improved/Must be improved/Not applicable |
|
|
Are all the cited references relevant to the research? |
Yes/Can be improved/Must be improved/Not applicable |
|
|
Is the research design appropriate? |
Yes/Can be improved/Must be improved/Not applicable |
|
|
Are the methods adequately described? |
Yes/Can be improved/Must be improved/Not applicable |
|
|
Are the results clearly presented? |
Yes/Can be improved/Must be improved/Not applicable |
|
|
Are the conclusions supported by the results? |
Yes/Can be improved/Must be improved/Not applicable |
|
|
3. Point-by-point response to Comments and Suggestions for Authors |
||
|
4. Response to Comments on the Quality of English Language |
||
|
Point 1: The English is fine and does not require any improvement. |
||
|
Response 1: We sincerely appreciate the reviewer's confirmation regarding the English quality of our manuscript. |
||
|
5. Additional clarifications |
||
|
[Here, mention any other clarifications you would like to provide to the journal editor/reviewer.] |
||

Reviewer 2 Report
Comments and Suggestions for Authors
I appreciate the authors' substantial efforts to revise the manuscript in response to prior feedback. The adoption of MaAsLin2, clarification of methodological steps, and revision of interpretative language mark clear improvements. However, I remain concerned about the robustness of the findings given the small sample size, marginal statistical significance of some results, and potential overfitting in the multi-omics model. I recommend further improvements of the conclusions, particularly regarding predictive modeling, and more detailed methodological clarification (e.g., normalization approach for MaAsLin2). With these adjustments, I believe the manuscript could be considered for publication.
Author Response
For research article
|
Response to Reviewer 2 Comments
|
||
|
1. Summary |
|
|
|
We sincerely thank the reviewer for the insightful comments that have strengthened our manuscript. We have carefully addressed concerns about sample size limitations, marginal significance, and potential overfitting through. We believe these revisions have substantially improved the manuscript's scientific rigor while maintaining its important findings. ]
|
||
|
2. Questions for General Evaluation |
Reviewer’s Evaluation |
Response and Revisions |
|
Does the introduction provide sufficient background and include all relevant references? |
Yes/Can be improved/Must be improved/Not applicable |
|
|
Are all the cited references relevant to the research? |
Yes/Can be improved/Must be improved/Not applicable |
|
|
Is the research design appropriate? |
Yes/Can be improved/Must be improved/Not applicable |
|
|
Are the methods adequately described? |
Yes/Can be improved/Must be improved/Not applicable |
|
|
Are the results clearly presented? |
Yes/Can be improved/Must be improved/Not applicable |
|
|
Are the conclusions supported by the results? |
Yes/Can be improved/Must be improved/Not applicable |
|
|
3. Point-by-point response to Comments and Suggestions for Authors |
||
|
Comments 1: I appreciate the authors' substantial efforts to revise the manuscript in response to prior feedback. The adoption of MaAsLin2, clarification of methodological steps, and revision of interpretative language mark clear improvements. However, I remain concerned about the robustness of the findings given the small sample size, marginal statistical significance of some results, and potential overfitting in the multi-omics model. I recommend further improvements of the conclusions, particularly regarding predictive modeling, and more detailed methodological clarification (e.g., normalization approach for MaAsLin2). With these adjustments, I believe the manuscript could be considered for publication.
|
||
|
Response 1: We sincerely appreciate the reviewer's insightful comments regarding the potential overfitting in the multi-omics model and more detailed methodological clarification in our study.
Key revisions made: 2. Materials and Methods (1)Normalization approach for MaAsLin2 Add: "MaAsLin2 analysis was performed using TSS (Total Sum Scaling) normalization with Log2 transformation to account for compositional data characteristics. Analysis method was performed using LM (Linear Model), Covariates (fixed_effects: age, sex, dietary) were included in the model to control for confounding effects, and multiple test corrections were performed using the Benjamini–Hochberg method (BH),associations with FDR <0.10 were considered exploratory given the small sample size and should be interpreted with caution. Additionally, we have implemented random effects in our computational approach to more accurately capture between-individual and between-group variations, thereby enhancing the reliability of our results.. " Location: Section 2.8.2, (Page: 9, Line: 354-359).
Added: Associations with FDR <0.10 were considered exploratory given the small sample size and should be interpreted with caution. Location: Section 2.8.2, (Page: 9, Line: 359-360).
(2) Overfitting in Random Forest Model Clarified: " the RF model was trained with 500 trees, and feature importance was validated using 10-fold cross-validation repeated 5 times to ensure stability. Permutation tests (n=1000) were performed to assess significance of AUC values." Location: Section 2.8.2, (Page: 8, Line: 327-330).
3. Results
(1)AUC Interpretation: Overstated predictive performance without external validation. Revised: "The high AUC (0.967) in our internal validation suggests potential discriminative power, but external cohorts are needed to confirm generalizability." Location: Section 3.2.4, (Page: 15-16, Line: 515,529-530).
(2) Marginal Significance Handling: Due to the small sample characteristics in this part of the study, we removed the differential bacterial genera with FDR values greater than 0.10, such as Phocaeicola(FDR=0.103) and Coprococcus (FDR=0.16), in order to provide a more cautious explanation. Meanwhile, the correlation analysis between the differential metabolites and the differential bacterial genera was re-conducted (marginal statistical significance was deleted). Revised: MaAsLin 2 analysis was performed to identify deferentially abundant genera across groups. In the association analysis of the OVOB, WAS, and NOR groups, a total of 12 differential bacterial genera were identified (FDR<0.10). As shown in Figure 11., compared with the NOR group, the OVOB group exhibited significant correlations with bacterial genera, including a negative correlations with SCFA-producing bacterial genera, such as Colidextribacter, Dysosmobacter, Intestinimonas,, Longicatena, and Butyricimonas (coefficients of -3.20, -3.03, -2.98, -1.04, and -2.92, respectively; FDR values of 0.02, 0.02, 0.02, 0.08, and 0.09 respectively). Alistipes were negatively correlated with children in the OVOB group (coefficients = -2.28, and FDR = 0.06). Associations with FDR <0.10 were considered exploratory given the small sample size and should be interpreted with caution. Location: Section 3.3.3, (Page: 19, Line: 591-601).
We deleted Parabacteroides (FDR=0.103) and Coprococcus (FDR=0.16) in the associaton of Section 3.4. Revised: In the OVOB group, significant correlations were observed between differential metabolites and microbial genera (Figure 12). PE (16:0/16:0) demonstrated positive correlations with multiple genera, including Butyricimonas (r = 0.448, P = 0.008), Dysosmobacter (r = 0.483, P = 0.003), Intestinimonas (r = 0.411, P = 0.015), Longicatena (r = 0.386, P = 0.024) and Alistipes (r = 0.422, P = 0.012). Methyl sorbate was positively associated with Butyricimonas (r = 0.432, P = 0.011) and Intestinimonas (r = 0.413, P = 0.015), whereas deacetylnomilin displayed negative correlations with Intestinimonas (r = -0.349, P = 0.043) and Alistipes (r = -0.399, P = 0.019). Location: Section 3.4, (Page: 20, Line: 609-616).
4. Discussion (1)Deleted the discussion section of the bacteria genera of FDR>0.10. (2)Added: Thus, Our exploratory findings nominate multi-omics signatures for further validation as potential stratification tools, pending replication in larger cohorts. Location: Section 4, (Page: 22, Line: 693-695).
5、Innovations and Limitations (1)Small Sample Size & Overfitting The microbiome sample size (n=42) may be underpowered to detect rare genera and increase overfitting risk, independent validation is essential. Location: Section 5, (Page: 23, Line: 704-705).
6. Conclusions We have adopted a more conservative approach in drawing our conclusions. Specifically: (1)For gut microbiota analysis, associations with FDR <0.10 were considered exploratory given the small sample size and should be interpreted with caution. (2)We have explicitly acknowledged the study's sample size limitations for gut microbiota analysis. (3)We have clarified the potential for overfitting in our Random Forest model analysis. (4)We emphasize the need for validation in larger cohorts. These adjustments provide appropriate scientific caution while maintaining the study's validity. Location: Section6, (Page: 23, Line: 709-721).
Revised manuscript: This multi-omics study characterized the preliminary metabolic and gut microbial profiles of Chinese children aged 5–7 years with obesity and metabolic heterogeneity phenotypes. Our data suggest that MUO children may exhibit alterations in ether lipid metabolism and sphingolipid metabolism pathways, showed tentative associations with elevated LysoPCs and reduced PEs/PIs. Obese children displayed reduced microbial diversity, an elevated Firmicutes/Bacteroidetes ratio, along with apparent reductions in certain SCFA-producing genera. Observed correlations between specific bacterial taxa and phospholipid metabolites could indicate microbial-host metabolic interactions, though alternative explanations including shared dietary influences cannot be excluded. These exploratory results contribute to the growing understanding of metabolic heterogeneity in childhood obesity but should be interpreted with appropriate caution until replicated in more diverse populations and supported by functional validation.
Thank you for this valuable suggestion, which has helped us present our findings more appropriately. |
||
|
4. Response to Comments on the Quality of English Language |
||
|
Point 1: |
||
|
We sincerely appreciate the reviewer's confirmation that our manuscript meets the journal's standards for English language quality. |
||
|
5. Additional clarifications |
||
|
[Here, mention any other clarifications you would like to provide to the journal editor/reviewer.] |
||
